# The Biological Role of Vitamins in Athletes’ Muscle, Heart and Microbiota

**DOI:** 10.3390/ijerph19031249

**Published:** 2022-01-23

**Authors:** Mariarita Brancaccio, Cristina Mennitti, Arturo Cesaro, Fabio Fimiani, Martina Vano, Biagio Gargiulo, Martina Caiazza, Federica Amodio, Iolanda Coto, Giovanni D’Alicandro, Cristina Mazzaccara, Barbara Lombardo, Raffaela Pero, Daniela Terracciano, Giuseppe Limongelli, Paolo Calabrò, Valeria D’Argenio, Giulia Frisso, Olga Scudiero

**Affiliations:** 1Department of Molecular Medicine and Medical Biotechnology, University of Naples Federico II, 80131 Naples, Italy; brancacciomariarita2@gmail.com (M.B.); cristinamennitti@libero.it (C.M.); martina.vano@gmail.com (M.V.); biagiogargiulo75@gmail.com (B.G.); iolanda.coto@unina.it (I.C.); cristina.mazzaccara@unina.it (C.M.); barbara.lombardo@unina.it (B.L.); pero@unina.it (R.P.); 2Department of Translational Medical Sciences, Università degli Studi della Campania “Luigi Vanvitelli”, 80138 Napoli, Italy; arturocesaro@hotmail.it (A.C.); amodio.federica@yahoo.it (F.A.); paolo.calabro@unicampania.it (P.C.); 3Division of Clinical Cardiology, A.O.R.N. “Sant’Anna e San Sebastiano”, 81100 Caserta, Italy; 4Unit of Inherited and Rare Cardiovascular Diseases, Azienda Ospedaliera di Rilievo Nazionale AORN Dei Colli, “V.Monaldi”, 80122 Naples, Italy; fimianifabio@hotmail.it; 5Inherited and Rare Cardiovascular Diseases, Department of Translational Medical Sciences, University of Campania “Luigi Vanvitelli”, Monaldi Hospital, 81100 Naples, Italy; martina.caiazza@yahoo.it; 6Department of Neuroscience and Rehabilitation, Center of Sports Medicine and Disability, AORN, Santobono-Pausillipon, 80122 Naples, Italy; ninodalicandro@libero.it; 7Ceinge Biotecnologie Avanzate S. C. a R. L., 80131 Naples, Italy; dargenio@ceinge.unina.it; 8Task Force on Microbiome Studies, University of Naples Federico II, 80100 Naples, Italy; 9Department of Translational Medical Sciences, University of Naples Federico II, 80131 Naples, Italy; daniela.terracciano@unina.it; 10Department of Cardio-Thoracic and Respiratory Sciences, Università degli Studi della Campania “Luigi Vanvitelli”, 80138 Napoli, Italy; limongelligiuseppe@libero.it; 11Department of Human Sciences and Quality of Life Promotion, San Raffaele Open University, Via di Val Cannuta 247, 00166 Roma, Italy

**Keywords:** micronutrients, athletic performance, nutrition, gut microbiota, cardiac pathologies, muscle damage

## Abstract

Physical activity, combined with adequate nutrition, is considered a protective factor against cardiovascular disease, musculoskeletal disorders, and intestinal dysbiosis. Achieving optimal performance requires a significantly high energy expenditure, which must be correctly supplied to avoid the occurrence of diseases such as muscle injuries, oxidative stress, and heart pathologies, and a decrease in physical performance during competition. Moreover, in sports activities, the replenishment of water, vitamins, and minerals consumed during training is essential for safeguarding athletes’ health. In this scenario, vitamins play a pivotal role in numerous metabolic reactions and some muscle biochemical adaptation processes induced by sports activity. Vitamins are introduced to the diet because the human body is unable to produce these micronutrients. The aim of this review is to highlight the fundamental role of vitamin supplementation in physical activity. Above all, we focus on the roles of vitamins A, B6, D, E, and K in the prevention and treatment of cardiovascular disorders, muscle injuries, and regulation of the microbiome.

## 1. Introduction

Vitamins are micronutrients of fundamental importance for our body [1], introduced daily with the diet, and found in nature [2,3,4,5,6].

Currently, 13 different types of vitamins have been described and classified according to their biological and chemical activity. Furthermore, these micronutrients can be divided into two large groups: water-soluble and fat-soluble vitamins [7,8].

Water-soluble vitamins are not stored in our bodies and generally require daily food intake. These are the B vitamins (B1, B2, B3, B5, B6, B8, B9, B12) and vitamin C [7], differentially distributed in vegetable and animal foods, milk, and its derivatives. By comparison, fat-soluble vitamins (A, D, E, and K) are absorbed together with dietary fats; they accumulate in the liver and adipose tissue, except for vitamin K, which is poorly stored in the tissues and therefore requires a continuous dietary supply [8]. Fat-soluble vitamins are found mainly in fruit and vegetables, with the exception of vitamin D, which is synthesized by the human body. Importantly, vitamins play a fundamental role in the regulation of many chemical reactions essential for life (Table 1) [9,10,11,12,13,14].

Hypovitaminosis is classified as primary or secondary vitamin deficiency. Primary hypovitaminosis occurs when the diet does not ensure an adequate vitamin intake or can be associated with particular eating styles (i.e., vegan nutrition). Secondary hypovitaminosis can be caused by a disease/condition that prevents or limits the absorption of vitamins. The lack of one or more vitamins can lead to severe and sometimes irreversible pathological conditions [29,30,31] (Table 2).

Conversely, the “side effects” caused by excessive vitamin intake are not common. However, high vitamin intake can cause some common ailments such as nausea, diarrhea, and vomiting. The elimination of these symptoms is often achieved through reducing vitamin dosage and also through consuming a balanced diet. The most important hypervitaminoses concern vitamins A, B6, C, D, E, and K [44,45,46] (Table 3).

In addition, it has been observed that vitamins have an essential role in COVID-19 and immunodeficiencies as they play an important role in activating the immune system and in stimulating lymphocytes [55,56].

Therefore, because physical activity is relevant for all ages [57,58], moderate exercise combined with satisfactory nutrition are protective factors against cardiovascular disease, muscular-skeletal disorders, obesity, diabetes, cancer, and cellular aging [57,58,59,60,61,62]. Moreover, it is well known that acute or strenuous exercise can lead to muscle injuries, cardiovascular maladaptation, and dysregulation of gut microbiota [63,64,65,66,67,68]. For these and other reasons, appropriate nutrition and adequate supplementation are necessary. In fact, vitamins A, B6, D, E, and K are essential for physical activity. 

In particular, vitamin A is involved in facilitating wound healing and maintaining the immune system against diseases and infections, especially those affecting the lungs [69]; moreover, vitamin A is one of the main micronutrients involved in the fight against oxygen free radicals. Competitive athletes are known to be at greater risk of antioxidant vitamin deficiency because of an increased use of O_2_. A marked deficiency can cause an increase in highly reactive oxygen molecules and therefore oxidative damage [70].

When the body is subjected to intense physical exercise, it is necessary that the metabolic processes ensure an adequate supply of energy to the tissues involved in the stress. In particular, two of these processes involve vitamin B6: the transformation of hepatic and muscular glycogen into glucose, and the deamination of amino acids to free carbon skeletons useful for gluconeogenesis. By comparison, Pyridoxal 5′-phasphate (PLP) represents the active form of vitamin B6 and is part of the enzyme glycogen phosphorylase; PLP is also the cofactor of aminotransferases, which degrade amino acids [71,72].

Vitamin D plays an important role in bone and skeletal health. The regulation of this vitamin in the human body is able to influence immune regulation and athletic performance [73]. Specifically, vitamin D upregulates gene expression of broad-spectrum antimicrobial peptides (AMPs), important regulators in innate immunity, and also downregulates expression of inflammatory cytokines such as tumor neurosis factor-alpha (TNF-alpha) and interleukin-6 (IL-6). Several proinflammatory cytokines, including IL-6 and TNF-alpha, are elevated after exercise, and increased levels have been associated with overreaching (overtraining) syndrome [74].

Vitamins E and A represent the main micronutrients having the role of antioxidants. In particular, vitamin E protects against the formation of reactive oxygen species (ROS) by promoting cell renewal [75]. An adequate antioxidant status may be important to maintain healthy muscle function, especially during the recovery phase after acute exercise and endurance exercise activities [76].

Finally, vitamin K is a leading player in the blood clotting process and ensures the functionality of the proteins involved in bone remodeling [77]. Low levels of this vitamin have been associated with increased bone turnover and fracture risk [78].

Therefore, this review focuses on the involvement of vitamins in human biological processes and, in particular, in the athletic population. Furthermore, it underlines how these micronutrients are essential in sports activities to support the maintenance of the gut microbiota and avoid cardiovascular and muscle injuries, thus safeguarding the athletic performance. 

## 2. Role and Function of Vitamins in Biological Processes

Generally speaking, vitamins act as coenzymes or prohormones by taking part in numerous human biological processes. It is known that vitamins must be taken with the diet; however, in some cases, our body is able to synthesize them. For convenience, we examine each vitamin in different sections to shed light on the processes in which each of them is involved.

### 2.1. Vitamin A

Vitamin A is the generic name of a mixture of vitamers, also known as retinoids (retinol, retinal, and retinoic acid), that show the biological activity of retinol (Figure 1). Sources of retinoids include animal products (meat, fish, eggs, and derivatives). Green, yellow, and orange plants contain carotenoids, which act as pro-vitamin A, giving rise to at least one molecule of retinol via enzymatic hydrolysis. β-carotene, specifically, contributes to the orange color of food and is typically associated with carrots and sweet potatoes.

Vitamin A belongs to a group of antioxidant vitamins—substances capable of neutralizing free radicals that are produced during intense physical exercise. Therefore, an adequate intake of vitamin A can contribute to the elimination of reactive oxygen species (ROS), and prevent the onset of diseases such as heart failure and muscle damage.

The liver plays a key role in vitamin A metabolism: retinol is esterified to retinyl esters and stored in the stellate cells [79]. The biological functions of retinoids are wide and varied and are related to different vitamers: retinal plays a regulatory role in the function of rod photoreceptors, which are responsible for black-and-white vision in low-intensity light; retinal participates in glycoprotein synthesis, acting as a cofactor in the transport of mannose to the protein component; and retinoic acid works as a transcriptional regulator. Therefore, retinoic acid intervenes in cell differentiation and morphogenesis, is essential for bone growth, guarantees the integrity of skin and mucous membranes, is necessary for reproductive function (regulation of spermatogenesis and embryogenesis), and regulates the differentiation of granulocytes from myeloid stem cells. Furthermore, it has a role in immunoregulatory processes [79]. Vitamin A has been found to regulate adaptive immunity, and promote T-lymphocyte differentiation and proliferation, especially of regulatory T-cells, memory B cells, and antibody-secreting plasma cells, particularly those involving IgA production. Finally, carotenoids have been shown to harbor antioxidative properties. 

According to the National Institutes of Health (NIH), the recommended dietary allowances (RDA) for vitamin A are 900 μg for adult males and 700 μg for adult females [5,80]. Hypervitaminosis A refers to the toxic effects of ingesting an excessive amount of preformed vitamin A [81]. Symptoms arise as a result of altered bone metabolism and altered metabolism of other fat-soluble vitamins [82]. Diagnosis is not easy because the serum retinol dosage is not sensitive to toxic levels of vitamin A, even if is the only effective test available. Hypervitaminosis A is usually treated by a drastic vitamin A reduction, resulting, in most cases, in full recovery. High intake of provitamin carotenoids (such as beta-carotene) from vegetables and fruits does not cause hypervitaminosis A, as conversion from carotenoids to the active form of vitamin A is regulated by the body to maintain an optimum level of the vitamin. Acute hypervitaminosis A may lead to headaches, nausea, vomiting, and anemia [83].

In contrast, hypovitaminosis A refers to poor vitamin A intake. Symptoms are recognized as an impaired dark adaptation of the eyes, which can lead at first to night blindness. In advanced deficiency, the cornea becomes hazy and can develop erosions, which can lead to its destruction (keratomalacia) [84]. Moreover, keratinization of the skin and of the mucous membranes in the respiratory, gastrointestinal, and urinary tracts can occur. The younger the patient, the more severe the effects of vitamin A deficiency. Furthermore, growth retardation and infections are common among newborns/children. The mortality rate can exceed 50% in newborns/children with severe vitamin A deficiency [85].

### 2.2. Vitamin D

Vitamin D (Figure 1) is a fat-soluble vitamin and, consequently, accumulates in the liver through food consumption. Moreover, it is released into circulation in small doses when its use becomes necessary. Vitamin D is also produced endogenously when ultraviolet (UV) rays from sunlight strike the skin and trigger vitamin D synthesis [71]. In fact, vitamin D comes in two forms: (1) ergocalciferol—which is found in plant foods; and (2) cholecalciferol—which is synthesized by our body or found in plant foods. Vitamin D-rich foods include cod liver oil, fatty fish (salmon, oysters, and shrimp), butter, egg yolk, mushrooms (the only vegetable source of vitamin D), and liver meat.

Moreover, vitamin D is a regulator of calcium metabolism and is therefore useful in calcifying bones and preventing hypocalcemic tetany (involuntary contraction of muscles, which leads to cramps and spasms) [86]. Vitamin D also contributes to maintaining normal levels of calcium and phosphorus in the blood. Furthermore, vitamin D has other roles in the body, including reducing inflammation and modulating processes such as cell growth, neuromuscular and immune functions [56], and glucose metabolism. Many genes encoding proteins that regulate cell proliferation, differentiation, and apoptosis are modulated in part by vitamin D. Moreover, many tissues have vitamin D receptors and some convert 25-hydroxy-vitamin D (25 (OH) D) to 1,25 dihydroxy-vitamin D (1,25 (OH) 2D).

The Food and Nutrition Board (FNB) committee determined that an RDA for vitamin D of 600 μg in adults equals the daily intake sufficient to maintain bone health and normal calcium metabolism in healthy people [87].

Hypovitaminosis D: A diet lacking in vitamin D in combination with inadequate sun exposure causes osteomalacia in adults and rickets in children, which consists of the rarefaction of bone tissue [37,38]. Today, in the Western world, these conditions are extremely rare. However, vitamin D deficiency has become a global problem for the elderly population, and remains common in children and adults in less developed countries. The low content of calcifediol (25-hydroxy-vitamin D) derives mainly from poor sun exposure, and deficiency leads to reduced bone mineralization and damage to the skeleton, leading to the aforementioned diseases [88].

Specifically, early indicators of vitamin D deficiency include serum reduction in calcium and phosphorus, secondary hyperparathyroidism, and an increase in serum alkaline phosphatase. Conversely, late indicators include inadequate mineralization of the skeleton, muscle weakness, and abdominal pain.

Hypervitaminosis D is caused by increased intestinal absorption and bone resorption of calcium, with consequent hypercalcemia, which is easily identifiable by the increase in urination and thirst [89].

If left untreated, hypercalcemia results in excess calcium deposits in soft tissues and organs such as the kidneys, liver, and heart, causing pain and organ damage. This is associated with a decrease in serum parathyroid hormone (PTH) and, finally, a loss of calcium homeostasis with severe manifestations such as anorexia, nausea, vomiting, diarrhea, hypercalcemia and hypercalciuria, nephrocalcinosis, cardiocalcinosis, and soft tissue calcification. These may be followed by polyuria, polydipsia, weakness, insomnia, nervousness, itching, and, eventually, kidney failure. In addition, proteinuria, urinary stones, BUN, and metastatic calcification (especially in the kidneys) may develop. Other symptoms of vitamin D toxicity include mental retardation in young children, abnormal bone growth and formation, diarrhea, irritability, weight loss, and severe depression [89].

According to recent studies, vitamin D deficiency reduces muscle function and strength, and can increase the risk of fractures due to stress and diseases that may have a detrimental effect on training and performance [74,90,91,92]. Conversely, elevated serum vitamin D levels in athletes have been associated with reduced injury rates and improved sports performance [74,90,91,92,93]. For these reasons, it is important to correctly identify people with vitamin D deficiency who require supplements to optimize their performance and prevent future injury.

### 2.3. Vitamin E

Vitamin E (Figure 1) is a fat-soluble compound, which consists of eight isoforms, four tocopherols (α-, β-, γ-, and δ-tocopherols), and four tocotrienols (α-, β-, γ-, and δ-tocotrienols), and is a lipid component of biological membranes. The various isoforms are not interchangeable and α-tocopherol represents the most biologically active compound [94]. 

The main source of vitamin E in the human diet varies depending on the isoform, with α-tocopherol found predominantly in foods such as nuts, almonds, hazelnuts, legumes, avocados, and sunflower seeds, and significant amounts are also available in green leafy vegetables and fortified cereals.

Vitamin E acts as a first-line defense against lipid peroxidation, protecting the cell membranes from free radical attack [75]. Moreover, vitamin E can inhibit lipid peroxidation by donating a hydrogen atom to peroxylipid radicals, thus making them less reactive. This redox reaction transforms vitamin E into an α-tocopheroxy radical, which is reactively stable, and which can react with vitamin C, glutathione, or coenzyme Q10 to reform α-tocopherol. Consequently, vitamin E is considered an important protective factor in all those processes in which negative effects from oxidative stress can occur; for example, diseases such as diabetes [95], cardiovascular and neurodegenerative diseases [96,97], and cancer [98], in addition to physiological conditions such as aging [99] or intense exercise [100].

The FNB recommends 15 mg of vitamin E per day in order to meet the body’s demands [101]. Vitamin E deficiency is rare in humans and occurs as a result of abnormalities in dietary fat absorption or metabolism. Mutations in genes coding for α-tocopherol transfer protein (α-TTP) represent an example of genetic abnormality in metabolism. In humans, this genetic defect is responsible for a progressive neurodegenerative disorder known as ataxia with vitamin E deficiency (AVED), despite consuming normal amounts of vitamin E [102]. Large amounts of α-tocopherol as a dietary supplement are needed to compensate for the lack of α-TTP. Vitamin E deficiency due to either malabsorption or metabolic anomaly can cause nerve problems.

In regards to vitamin E excess, no adverse events related to high levels of vitamin E have been reported in the literature [5]. However, it has been observed that high doses of α-tocopherol supplements can cause bleeding and stop blood clotting in animal models, whereas high doses are responsible for inhibiting platelet aggregation. 

In recent years, several research groups have focused their attention on the use of vitamin E in sports. In fact, it has been found that endurance exercise can cause an increase in oxidative low-density lipoprotein (LDL) concentrations and predispose athletes to an increased risk of developing atherosclerosis. However, high levels of oxidized LDL can be reduced if vitamin E levels are kept high [97]. Therefore, the correct intake of vitamin E in athletes represents rational support in countering the onset of adverse events such as atherosclerosis.

### 2.4. Vitamin K

Vitamin K (Figure 1), also known as naphthoquinone, belongs to the group of fat-soluble vitamins and, as such, must not be continuously consumed via food intake. In fact, vitamin K is stored in the body and used when required. Vitamin K is essential for the hepatic synthesis of prothrombin and other blood-clotting factors [27]. It also plays an important role in ensuring the functionality of the proteins that maintain bone health [103].

Vitamin K is divided into three groups based on its origin and functions: vitamin K1 is of vegetable origin and participates in the blood coagulation processes; vitamin K2 (also known as “menaquinone”) is of bacterial origin, promotes the absorption of intestinal microflora, and is essential for bone well-being; and vitamin K3 (“water-soluble menadione”) is of synthetic origin and is included in those drugs whose function is to regulate blood coagulation processes [104]. Vitamin K is mainly present in vegetables (tomatoes, spinach, cabbage, turnip greens) [105], whereas it is lacking in foods of animal origin (with the exception of animal liver). It is also produced by our intestinal bacterial flora.

The FNB has established an adequate intake (AI) of vitamin K in the healthy population. In adults, this corresponds to 120 mg for males and 90 mg for females [106]. Regarding vitamin K deficiency, this condition determines a propensity to bleed as it is essential for the hepatic synthesis of prothrombin, proconvertin, and other substances involved in blood clotting [27,41]. Vitamin K deficiency is a rare event and is linked above all to pathologies that prevent regular intestinal absorption or prolonged antibiotic treatment. Vitamin K deficiency also causes problems affecting bones and joints with fractures, osteoporosis, and various forms of osteoarthritis [107,108]. Conversely, in vitamin K excess, which rarely occurs in adults, there is an increased vitamin K levels In this case, symptoms include thrombosis, vomiting, anemia, excessive sweating, hot flashes, tightness in the chest, or naphthoquinone, a condition that occurs when excessive doses of vitamin supplements are consumed [54].

In addition, vitamin K is often considered a nutrient for improving heart health, reducing cancer risk, and increasing bone density [109]; however, it also appears to improve fitness, even in healthy athletes [110,111]. Consequently, like most nutrients, it appears to have quite versatile roles.

### 2.5. Vitamin B6

Vitamin B6 (Figure 1) is among the so-called water-soluble vitamins, which cannot be accumulated in the body, and is therefore regularly consumed through food. The term vitamin B6 refers to a group of substances that include pyridoxine, pyridoxal, pyridoxamine, and their respective 5′-phosphate esters. Pyridoxal 5′-phosphate (PLP) and pyridoxamine 5′-phosphate (PMP) are active coenzyme forms of vitamin B6. PLP acts as a coenzyme in transamination reactions, and in certain decarboxylation, deamination, and racemization reactions of amino acids [112]. Therefore, vitamin B6 is involved in the metabolism of amino acids, lipids, and carbohydrates [113]. In addition, it contributes to cognitive development, immune function, and hemoglobin formation [114].

The richest sources of vitamin B6 include fish, beef liver and other organs, potatoes, and other vegetables rich in starch and fruit. 

The FNB established the recommended daily allowance (RDA) of vitamin B6 in both men and women, corresponding to 1.3 mg of vitamin B6 [115]. Vitamin B6 deficiencies are rare. Moreover, low levels of vitamin B6 are found in chronic alcohol dependence, pregnancy, preeclampsia and eclampsia, and malabsorptive states such as celiac and inflammatory bowel disease [114]. Symptoms include microcytic anemia, electroencephalographic abnormalities, dermatitis, depression and confusion, and weakened immune function. In contrast, regarding vitamin B6 excess, high levels of vitamin B6 are not associated with adverse effects. However, chronic administration of 1–6 g of oral pyridoxine daily for 12–40 months can cause severe and progressive sensory neuropathy characterized by ataxia. The severity of symptoms depends on the dose and symptoms resolve upon discontinuation of administration. Other effects from excessive vitamin B6 intake include painful and disfiguring skin lesions, photosensitivity, and gastrointestinal symptoms, such as nausea and heartburn [116].

Athletes, especially women, can be affected by vitamin B6 deficiency due to improper nutrition [117]. Vitamin B6 is an essential micronutrient in homocysteine amino acid metabolism, and low levels of vitamin B6 have been associated with hyperhomocysteinemia [118]. Increased homocysteine levels in the blood can cause heart and vascular disease, bone fragility, and neurodegenerative disease. Consequently, it would be useful for athletes to introduce an adequate amount of vitamin B into their diet in order to avoid exposure to these risks.

## 3. Crosstalk between Vitamins and Gut Microbiota in Athletes 

The intestinal microbiota plays a fundamental role in promoting and maintaining our health by actively influencing food metabolism, vitamin synthesis, energy production, inflammatory, and immune responses [119] (Figure 2). Moreover, factors such as age, diet, and drugs actively affect the composition of the microbiota. To these, physical exercise has recently been added with related psycho-physical stress components (specific diet, environment, performance, training) especially if practiced at a competitive level [120,121]. Following these considerations, we decided to consider vitamin’s involvement in maintenance of a healthy gut microbiota and, as a consequence, we chose to highlight the importance of gut microbiota in athlete’s performance. 

Given the recent attention to this aspect, many factors still need to be explored on how physical activity precisely affects the bacterial component. However, it is important to address this issue in order to optimize athletes’ performance and recovery.

The intestinal microbiome contains several bacterial phylotypes; the most represented are *Bacteroides* and *Firmicutes* (representing about 90% of microbial species). These microorganisms play an active and important role in numerous physiological processes [122,123].

Therefore, it is essential to maintain an optimal balance (defined as eubiosis) between the body and the ecosystem. Furthermore, it is now known that the microbiota provides the host with a series of metabolic, immunological, and protective functions (Figure 2).

From a metabolic function point of view, the microbiota promotes digestion and absorption of nutrients, and provides metabolites that are important for our health (folate, vitamin K2, and short-chain fatty acids (SCFA)). Partially digested foods, and in particular fibers, are fermented by the microbiota into SCFAs, i.e., butyrate, acetate, and propionate, producing numerous compounds with hormonal and/or neurotransmitter-like activity that are released into the bloodstream and can thus go on to exert their activity on different tissues, e.g., to modulate intestinal motility. Furthermore, physical activity has been shown to modulate the synthesis of SCFA, exerting a particular influence on butyrate synthesis [123].

Furthermore, numerous studies have investigated the effect of vitamins on gut microbiota. In this scenario, vitamin A plays a crucial role in determining the differentiation of B cells into the IgA secreting-cells present in the gut mucosa. These cells interact with the gut microbiota and this process is crucial for tolerance acquisition. Gommerman et al. have shown that a reduced vitamin A or a bacterial dysbiosis reduce IgA production, causing malabsorption and alterations in body fat deposition [124]. Vitamin D has been reported to influence the intestinal immunity by promoting the release of antimicrobial peptides, phagocytic activity, the expression of tight junction proteins, and lymphocytes’ differentiation into the anti-inflammatory Th2 cells [125]. At the same time, *Bashir* and colleagues observed that an 8-week oral supplementation with vitamin D3 is associated with an increase of CD8+ T cells and specific modifications of the gut microbiota, i.e., a reduction in Proteobacteria and an increase in Bacterioidates [126]. Instead, Selhub et al. investigated, in a mouse model of inflammatory bowel disease, the effects of dietary vitamin B6 supplementation on colonic inflammation. Interestingly, they found that a moderate supplementation and a light depletion were both able to ameliorate the inflammatory clues [127]. Little is known about the influence of dietary vitamin K on gut microbial composition; however, Ellis et al. showed that male and female mice, subjected to a diet low in vitamin K, had an alteration of the gut microbiome [128].

By comparison, athletic components such as exercise, associated with dietary factors, promotes a more “health-associated” gut microbiota (Figure 2). Typical features include a higher abundance of health-promoting bacterial species, increased microbial diversity, functional pathways, and stimulation of bacterial abundance that can modulate mucosal immunity, and improved barrier functions (Figure 2). In comparison to sedentary controls, athletes have increased fecal metabolites and improved overall health. It is also true that athletes generally have a distinct diet where the intake of vitamins is very important to reduce deleterious effects of overtraining and improve the composition of the microbiome [68]. In fact, physical activity has been shown to be able to modulate the synthesis of SCFAs, exerting a particular influence on the synthesis of butyrate [123].

In their study, Campbell and Wisniewski compared a group of rugby players with sedentary controls. First, questionnaires were collected on the physical activity performed and the diet followed, and finally, fecal samples were analyzed. Although there were differences in diet, it was found that rugby players had greater microbial diversity, synonymous with a “healthier” microbiota [123].

Petersen et al., comparing the microbiota of 22 professional cyclists vs. 11 amateurs, showed that exercise level is positively associated with the abundance of *Prevotella*—a genus related to the metabolism of amino acids and carbohydrates. Moreover, they observed an increased expression of *Akkermansia*, which is generally associated with a healthy microbiota, with, in addition, a lower presence of *Bacteroides* [12].

O’Donovan et al., comparing the microbiota of 37 professional Irish athletes playing 16 different sports, evaluated the impact of dynamic vs. static athletes. They observed that the microbiota composition was influenced by the type of sport. Indeed, there was significant variability between dynamic and static sports. Higher levels of lactate in the urine were then recorded in athletes performing static sports, and creatinine in the stools of athletes performing dynamic sports [125].

Another study compared premenopausal women: a sedentary group and a group practicing moderate physical activity. The diets followed by women were also well characterized in these studies. However, the physical activity carried out did not affect the microbiota diversity. Further analyses conducted in these women, however, showed that in the physically active group there was a greater presence of *R. hominis, A. municiphila*, and *F. prausnitzii*—these species are related to improved intestinal health [4]. Concurrently, the authors determined the enzymatic activity in the fecal samples of the two groups. This analysis showed that bacterial cysteine aminopeptidase activity was increased (6.6 times) in active women compared to sedentary women. Normally, the activity of this enzyme indicates the presence of *lactobacilli*; however, despite the activity, *lactobacillus* was not detected in the two study groups [126].

In addition, Bermon et al. compared the microbiota of young adults following a similar diet but with varying degrees of training [127]. The degree of training was assessed by measuring VO_2_max. The study showed that about 20% of the variation in alpha-diversity (inter-individual diversity) may have been due to VO_2_max. Furthermore, cardiorespiratory capacity has been shown to be a good indicator of intestinal microbial diversity. The study showed important metabolic changes, with a decrease in LPS synthesis pathways and an increase in butyrate pathways [127]. Adaptations to long-lasting physical exercises may therefore lead to changes in the GI tract that can affect the intestinal ecosystem (pH, nutrient absorption capacity, etc.). Consequently, we can also conclude that the microbiota, and its associated metabolism, is positively adapted to the more physically active lifestyle of the subjects undertaking the most training [127].

Finally, a study by Derman and Lambert showed that athletes are often affected by chronic fatigue syndrome, a pathology that is common yet difficult to diagnose by sports doctors [128]. Nevertheless, the fatigue condition may be common for any athlete during periods of high-volume training. Physicians must be able to distinguish between this physiological fatigue and more prolonged and severe fatigue, which may be due to a pathological condition. It was found that intestinal dysbiosis plays a key role in the manifestation and severity of this syndrome [129]. The analysis of the intestinal microbiota in patients affected by this disorder revealed, compared to healthy subjects, a lower diversity between bacterial species, an increase in pro-inflammatory species, such as *Proteobacteria* and *Prevotella*, and a decrease in anti-inflammatory species, in particular, *Faecalibacterium prausnitzii* and *Bifidobacteria*. This microbiota composition is probably caused by a diet that is vitamin and probiotic poor relative to the workload endured by the athlete’s body. In fact, it has been shown that patients affected by the fatigue syndrome often have vitamin deficiencies—depending on the lack of vitamin E, B, and K [129,130,131]. The increase in pro-inflammatory bacterial species most likely causes a decrease in probiotic bacterial species (particularly *bifidobacteria* and lactic acid bacteria), which are able to synthesize vitamin K and various B vitamins, such as cobalamin (vitamin B9), folic acid (vitamin B12), thiamine (vitamin B1), pyridoxine (vitamin B6), and riboflavin (vitamin B2). Therapeutic approaches aimed at modifying the gut microbiota composition may be a potential tool for controlling the development and/or progression of the syndrome [132,133].

Although further and appropriate studies are needed, especially in humans, this preliminary information represents a good starting point for establishing therapeutic interventions mediated by the microbiome, preventive or supportive health, and athletic performance.

## 4. Vitamin Deficiency of Athletes’ Cardiac Disorders 

In athletes, nutrient deficiency can negatively regulate the body’s repair activity (Figure 3). Micronutrients play an important role in energy metabolism, hemoglobin synthesis, maintenance of bone health, and stimulation of the immune system. Therefore, increasing the intake of micronutrients may be required to support the building, repair, and maintenance of lean body mass in athletes. Vitamins that have important roles in the athlete’s diet include vitamin D, B vitamins, and vitamins C, E, and K. Several research groups have established associations between vitamins and cardiac structure and function in human and animal models, specifically focusing on vitamin D deficiency [134]. In the literature, it is known that most of the world’s population is vitamin D deficient [135]. Furthermore, there is a known correlation between vitamin D deficiency and sudden cardiac death, especially in athletes [136]. Heart disease is considered a serious danger for athletes who undertake intense and prolonged competitive activity. For this reason, athletes are routinely monitored by sports physicians, who collect biomedical and personal data [137]. 

Several studies have demonstrated that cardiovascular diseases are accompanied by an increased expression of the vitamin D receptor. Vitamin D receptors (VDR) are present in the heart and vascular system, located specifically in cardiac myocytes and fibroblasts [138,139]. Therefore, vitamin D supplementation in people with vitamin D deficiency can facilitate the implementation of VDRs, which leads to structural remodeling of the heart muscles, vascular tissue, and activation of myocyte contractility [140]. VDR activation plays a role in cardiomyocyte proliferation, resulting in a beneficial adaptation mechanism [141]. Moreover, vitamin D downregulates genes that are involved in the development of myocardial hypertrophy [141]. 

In athletes, vitamin D deficiency causes long-term adverse cardiovascular effects on cardiac contractility, vascular tone, cardiac collagen content, and the maturation of cardiac tissue. These negative effects are the consequence of increases in the parathyroid hormone (PTH) level, which can lead to left ventricular hypertrophy (Figure 3). Hypertrophy alters the filling capacity of the ventricle and the ejection fraction, causing hypoxia of the muscle tissue, which decreases athletic performance [142,143]. These factors are critical to the performance and stamina of aerobic and anaerobic exercises [144]. Vitamin D supplementation in patients with severe deficiency results in an improvement in heart muscle function versus control cases. Vitamin D receptors are present at the vascular level, suggesting an influence on vascular physiology and pathophysiology. Furthermore, vitamin D deficiency is related to increased arterial stiffness and endothelial dysfunction in blood vessels, thus promoting atherogenesis. 

This confluence of alterations supports the hypothesis that inadequate vitamin D levels negatively affect cardiorespiratory capacity, resulting in variations in oxygen and nutrient supply to the exercised muscle.

Regular intensive physical exercise often leads to structural and electrophysiological cardiac adaptations. These modifications determine increased diastolic filling and increased cardiac output. This final enhancement is necessary to achieve athletic excellence. The entire set of modifications is known as “athlete’s heart” [144]. Importantly, many parameters play a significant role in the modification of the athlete’s heart, such as sex, ethnicity, age, duration, and intensity of the exercise. 

Allison et al. analyzed four groups of athletes with different levels of vitamin D: sufficiency (>30 ng/mL), insufficiency (20–30 ng/mL), deficiency (10–20 ng/mL), and severe deficiency (<10 ng/mL) [145]. Severely 25 (OH)D-deficient athletes presented significantly (*p* < 0.05) smaller aortic root and left atria diameters, intraventricular septum diameter (IVSd), left ventricular diameter during diastole (LVIDd), left ventricular mass (LVM), left ventricular volume during diastole (LVvolD), and right atrial (RA) area than insufficient and sufficient athletes. Allison et al. used logarithmic transformation to evaluate vitamin D level effects in athletes and in control participants. They adjusted values for age, ethnicity, body surface area, and athletic effort level, and focused on the positive association between vitamin D serum level and IVSd, LVIDd, left ventricle by pulsed-wave tissue Doppler (PWTd), LVM, LVvolD, and athletic preparation. This association is conversely absent in sedentary participants. 

Therefore, several studies have highlighted that athletes with severe vitamin D deficiencies have significantly smaller cardiac structural parameters than athletes with insufficient or sufficient vitamin D concentration. 

Recently, vitamin K deficiency has been associated with an increased risk of cardiovascular disease (Figure 3). Vitamin K deficiency results in inactive Gla proteins, resulting in an increased risk of vascular calcification and cardiovascular disease [146]. On this basis, McFarlin et al. investigated the potential role of vitamin K as a promoter of cardiovascular health in athletes. Strenuous exercise may have a negative impact on the cardiovascular system, resulting in negative cardiac output, which in turn may affect athletic performance. For this study, 26 male and female athletes were recruited and underwent placebo or vitamin K2 supplementation for eight weeks. The data showed that vitamin K2 supplementation was associated with significative improvements in cardiac output, and there was a trend towards changes in heart rate and stroke volume compared to the control group [147].

## 5. The Influence of Vitamins on Athletes’ Muscle Damage

Various sports practices are characterized by intense and prolonged physical commitment; triathlons, ultra-marathons, and other high intensity and duration sports can be responsible for muscle damage (Figure 4). The extent of damage depends on several parameters, such as the degree of training, genetics, age, intensity of exertion, and state of hydration. In this context, researchers are questioning the treatments to which athletes should be prescribed [148] and the influence that the diet, in particular the administration of micronutrients such as vitamins, have on muscle damage and whether these supplements may have an influence on athletic performance (Figure 4).

At the muscle level, vitamin D exerts its function by binding to nuclear or membrane receptors through genomic and non-genomic mechanisms [149]. The former, which is activated by binding to the nuclear receptor, regulates the proliferation of muscle cells and muscle growth. By comparison, the non-genomic mechanism involves binding to a membrane receptor, resulting in an influx of calcium ions into the cell, regulation of intra and extracellular ion levels, homeostasis of compounds containing phosphorus, and stimulation of PTH secretion. This may be responsible for an increase in strength, force, and muscle contraction. Furthermore, vitamin D inhibits the expression of myostatin, a negative regulator of muscle mass [26,139]. Finally, vitamin D is involved in the proliferation and differentiation of muscle cells, and in the inhibition of their apoptosis, establishing a link with athletic performance. In fact, vitamin D deficiency can cause myopathy, reduce muscle tone, and lead to increased degradation of type II muscle fibers, with negative consequences for muscle strength and power (Figure 4). 

Although vitamin D is usually associated with bone homeostasis, it has also been shown to be involved in calcium homeostasis [114]. Under physiological conditions, the mitochondria of skeletal muscle fibers absorb the cytoplasmic calcium released by the sarcoplasmic reticulum during contraction. Vitamin D deficiency reduces calcium re-uptake in the sarcoplasmic reticulum, lengthening the relaxation phase of muscle contraction [150]. Therefore, vitamin D deficiency is responsible for altering mitochondrial function and, consequently, for causing alterations in cellular metabolic homeostasis [151]. Clinical studies have shown that low vitamin D levels can induce high oxidative stress and alter the activity of antioxidant enzymes, thus increasing reactive oxygen species (ROS) levels. If vitamin deficiency persists for a long time, the function of the vitamin D receptor (VDR) in the muscle will be altered, with the consequent formation of ROS and altered mitochondrial function. In turn, this causes muscle atrophy, defined as a decrease in the size of a tissue or organ due to cell shrinkage. Muscle wasting results in an altered balance in the rates of protein degradation and protein synthesis [152].

It has been well demonstrated that conditions of hypovitaminosis D are more frequent in athletes than in the general population (Figure 4). The risk of hypovitaminosis D is strictly related to the latitude, time of year (winter and spring), and type of sport practiced (indoors vs. outdoors). Therefore, several studies have investigated the effects of vitamin D administration on serum cholecalciferol concentrations and physical performance [152,153]. The results showed that, although the athletes achieved a sufficient level of vitamin D, the effect on performance was not significant. Therefore, further studies are needed to investigate this aspect.

In recent years, it has been shown that intense exercise can lead to alterations in the oxidant/antioxidant balance [154], resulting in excessive ROS accumulation. In skeletal muscle, the increase in mitochondrial electron carrier activity, low catalase concentrations, increased oxygen supply, and consumption are responsible for the excessive accumulation of ROS, which in turn can cause skeletal muscle injury [155,156]. Moreover, the accumulation of ROS can cause damage to lipid membranes, a phenomenon known as lipid peroxidation, in which the deterioration of polyunsaturated fats occurs [157]. Vitamin E belongs to the non-enzymatic antioxidant defense system (Figure 4), whose main function is to interrupt lipid peroxidation reactions and remove free oxygen radicals, stabilizing cell membranes [158]. Because the antioxidant mechanisms implemented by the body are not always sufficient to counteract pro-oxidant events [159], some researchers have investigated the effects of possible vitamin E supplementation. A study by Meydani et al. showed that supplementation of 800 mg for 30 days contributed to an increase in plasma concentrations of α-tocopherol [160,161]. Other studies have obtained similar results, highlighting that vitamin intake has no significant effects on lipid peroxidation and that further investigations are needed to evaluate this effect.

Vitamin B6 represents an important factor in metabolic pathways that are involved in physical exercise, especially amino acid metabolism, and activates the rate-limiting step of glycogen breakdown [162].

Vitamin B6 is involved in homocysteine metabolism, and vitamin B6 deficiency is the primary cause of hyperhomocysteinemia and homocystinuria [163]. Elevated homocysteine levels have been associated with reduced muscle function [164]. Vascular inflammation, thrombosis, and thrombo-embolism are deleterious consequences of hyperhomocysteinemia and result in peripheral arterial disease (PAD), which is responsible for muscular damage, inflammation, and loss of muscle regeneration capability [165]. In addition, in muscle, efficient blood levels regulation and vascular system integrity ensure muscular endurance and adaptability to various external stimuli. The blood levels in muscle cells are typically regulated by nitric oxide (NO), synthesized by nitric oxide synthase (NOS). Mislocalization of NOS from sarcolemma and defective NO production were reported in several forms of muscular dystrophies, and to be involved in focal ischemia, diminished exercise endurance, and fatigue. On this basis, excess homocysteine may compromise NO signaling, causing fatigue, ischemia, and reduced physical resistance. Some studies have reported that athletes, especially female athletes, consume less vitamin B6 than recommended, thus exposing themselves to reduced performance and a higher risk of fatigue or injuries (Figure 4) [166,167,168,169,170,171,172]. 

In this scenario, the correct intake of the analyzed micronutrients may represent a valid tool to combat the increased risk of muscle damage and disorders, which can help in avoiding any loss of form and performance in athletes.

## 6. Conclusions

The importance of an adequate vitamin intake, given the involvement of these micronutrients in numerous human biological processes, is now well known. A deficit in their levels can cause severe pathologies; conversely, hypervitaminosis is almost exclusively a result of the excessive ingestion of vitamin supplements, rather than a degenerative physiological process. 

The data collected in this review suggest that a vitamin-rich diet accompanied by controlled physical activity can protect the body against the onset of serious pathophysiological disorders [173,174]. Furthermore, the intake of food supplements, especially in sports activity and in elite athletes, can help prevent injuries and improve performance. A healthy lifestyle is necessary to counteract the appearance of premature cell aging, heart disease, dysbiosis, and muscle damage, which is why it is recommended for all age groups. Further studies are needed to support this evidence.

In the near future, we hope that clinical studies will provide the necessary evidence to shed light on the clinical evaluation of vitamins, corresponding to their antioxidant, antimicrobial, and immunomodulatory properties.

## Figures and Tables

**Figure 1 ijerph-19-01249-f001:**
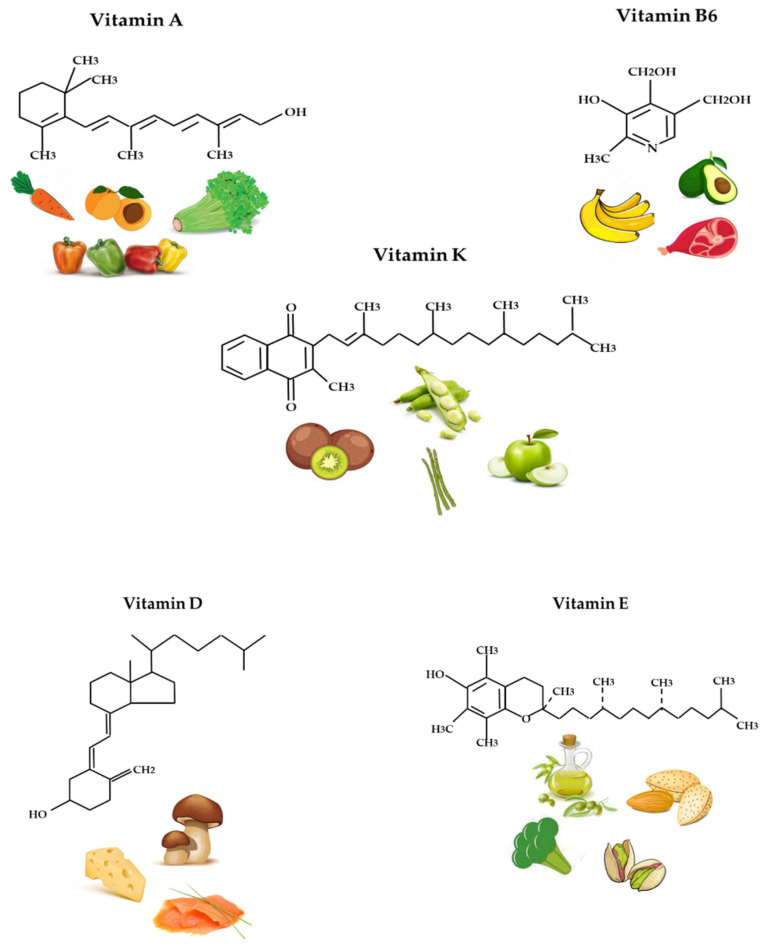
Chemical structures and food sources of vitamin A, B6, K, D, and E.

**Figure 2 ijerph-19-01249-f002:**
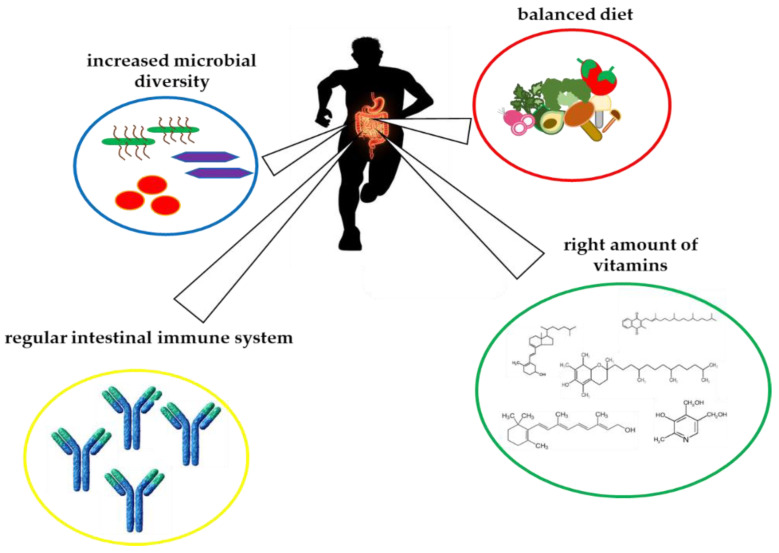
Schematic representation of the interplay between vitamins and athletes’ gut microbiota.

**Figure 3 ijerph-19-01249-f003:**
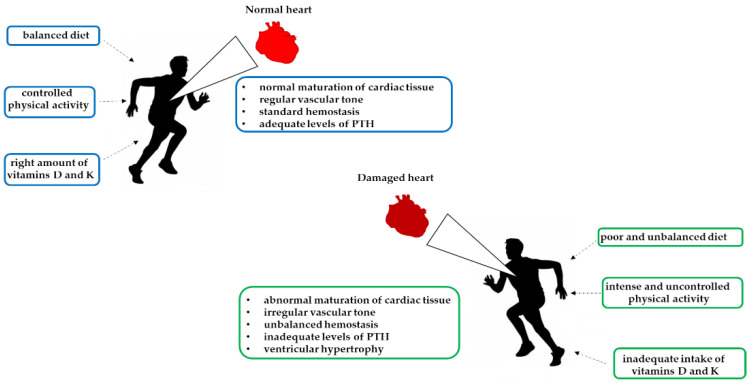
Schematic representation of hypovitaminosis effects and strenuous exercise on the heart.

**Figure 4 ijerph-19-01249-f004:**
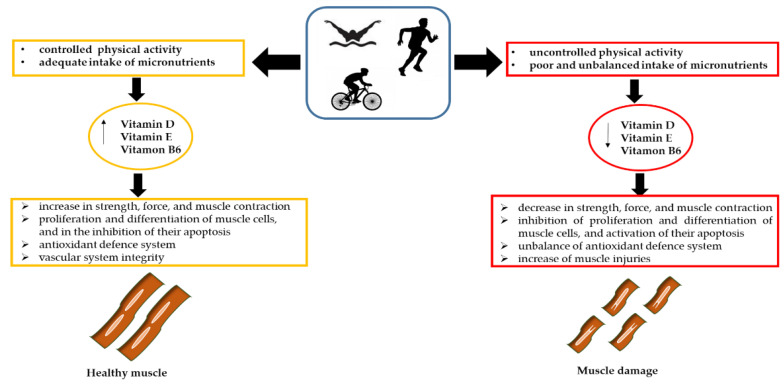
The role of vitamins in athletes’ muscle damage.

**Table 1 ijerph-19-01249-t001:** Physiological functions of vitamins in humans.

Micronutrient	Physiological Functions	References
Vitamin A	Improvement of vision, antioxidant, maintenance of the immune system, maintenance of healthy skin.	[15,16]
Vitamin B	Energy production, nucleic acid, protein, sugar and fat metabolism, maintenance of the immune system, psychological functions.	[17,18]
Vitamin C	Antioxidant, wound healing, maintenance of the immune system, maintenance of healthy skin, teeth, and gums.	[19,20,21]
Vitamin D	Healthy bones and tissues, modulation of cell growth, maintenance of the immune system, teeth growth.	[22,23,24]
Vitamin E	Antioxidant, maintenance of the immune system, prevention of cardiovascular diseases, vision protection.	[20,25]
Vitamin K	Blood clotting, bone strengthening, cardiovascular disease prevention, antioxidant.	[26,27,28]

**Table 2 ijerph-19-01249-t002:** Vitamin deficiency disorders in humans.

Micronutrient	Disorders	References
Vitamin A	Night blindness, growth disturbance, dysfunctions to the reproductive system; dysfunctions of the immune response.	[32,33,34]
Vitamin C	Scurvy, connective tissue disorders.	[35,36]
Vitamin D	Rickets, osteomalacia.	[37,38]
Vitamin E	Muscle metabolism disorders, neuropathy, oxidative hemolysis.	[39,40]
Vitamin K	Delayed coagulation, Hemorrhagic disease of the newborn.	[41]
Vitamin B6	Dermatitis, polyneuritis, muscle spasms.	[39,42,43]

**Table 3 ijerph-19-01249-t003:** Hypervitaminosis effects in humans.

Micronutrient	Diseases	References
Vitamin A	Headache, vomiting and numbness, anemia, teratogen in the fetus.	[39,47]
Vitamin D	Vomiting, headache, diarrhea, polyuria, calcinosis, fatigue.	[48,49]
Vitamin B6	Damage to the nervous system.	[50]
Vitamin C	Kidney stones, intestinal disorders.	[51,52]
Vitamin E	Absorption reduction in other liposoluble vitamins.	[53]
Vitamin K	Anemia, vomiting, thrombosis, excessive sweating.	[54]

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
