# Peer review of "The Biological Role of Vitamins in Athletes’ Muscle, Heart and Microbiota"

_ijerph, 2022, doi:10.3390/ijerph19031249_

Round 1
Reviewer 1 Report
This manuscript is much improved from the first version, especially regarding the microbiome. This is a welcome addition as much of the other information is not new. The main concern is the grammar and sentence structure which needs a great deal of work or future readers will be confused. Below are a few examples from the abstract only:
Line 35: should be ..... are considered protective factors .....
Line 37: .... must be correctly supplied ....
Line 38: What does shape of performance mean?
Line 39 .Moreover in sport activity is essential ...? Is something omitted here?
Line 53: Are there 12 or 13 vitamins? Here you indicate 12 and in lines 57-59 you list 13!
As indicated, the above are only a few examples. The entire manuscript needs considerable work with respect to grammar / sentence structure. Also, please eliminate one sentence paragraphs.
Once this extensive editing is done, I am willing to re-evaluate the manuscript. Until then is is not possible to make a recommendation.
Author Response
Dear Editor,
thank you for the Reviewer’s Report about our manuscript entitled “The biological role of vitamins in athlete ‘s muscle, heart and microbioma”, submitted to International Journal of Environmental Research and Public Health (IJERPH) in the Special Issue "Healthy Nutrition and Physical Activity". We have appreciated the comments received by the Reviewer and yourself and have carefully re-considered them in preparing a new version of the manuscript.
A point-by-point response to the comments is attached below.
We believe that the manuscript is now significantly improved thanks to the Reviewer’s inputs.
We hope that the new version of the paper deserve publication on International Journal of Environmental Research and Public Health.
Best regards,
Prof. Dr. Olga Scudiero and Prof. Dr.Giulia Frisso
Point-by-point response.
Reviewer 1
This manuscript is much improved from the first version, especially regarding the microbiome. This is a welcome addition as much of the other information is not new. The main concern is the grammar and sentence structure which needs a great deal of work or future readers will be confused. Below are a few examples from the abstract only:
Line 35: should be ..... are considered protective factors .....
Line 37: .... must be correctly supplied ....
Line 38: What does shape of performance mean?
Line 39 .Moreover in sport activity is essential ...? Is something omitted here?
Line 53: Are there 12 or 13 vitamins? Here you indicate 12 and in lines 57-59 you list 13!
As indicated, the above are only a few examples. The entire manuscript needs considerable work with respect to grammar / sentence structure. Also, please eliminate one sentence paragraphs.
Once this extensive editing is done, I am willing to re-evaluate the manuscript. Until then is is not possible to make a recommendation.
Thanks for the suggestions, in accordance with the recommendations, we have made corrections and we have improved the grammar using the English editing service of MDPI.
Grammar corrections are shown in track changes and changes in yellow.
Reviewer 2
The suggestions and corrections were accepted and the manuscript has been improved as suggested.
Thank you, we are happy to have satisfied your requests.
Reviewer 3
The authors attempt to review the role of vitamin nutrition on muscle and heart health and the gut microbiota in athletes. Comments:
The manuscript requires extensive editing for error in language. Even the title contains errors.
The beginning overview section is long and contains basic information that seems unnecessary for a journal publication.
The microbiota section contains no insights about vitamin status and microbiota abundance and diversity. The effect of exercise on microbiota parameters has been shown in multiple published studies and systematic reviews.
Many of the proposed links between vitamin status and exercise outcomes are discussed without direct evidence of a causal relationship.
The manuscript overall does not provide valuable new insights that can be implemented in the field of sports nutrition.
Thanks for the suggestions, in accordance with the recommendations, we have made corrections and we have improved the grammar using the English editing service of MDPI.
Grammar corrections are shown in track changes and changes in yellow.
Academic Editor Notes
A great implementation of description and discussion of topic is necessary. The authors should mark the novelty character of this paper respect to previous ones. I suggest to change the type of paper into "Perspective"
Thanks for the suggestions, in this latest version we have completed the corrections suggested by the reviewers. In particular, the first reviewer suggested that we shorten the introduction and carry out a profound grammar correction, the second was very satisfied and finally the third asked us to improve and further adjust the paragraph on the microbiome and grammar.
On our part, there was a commitment to modify and accept the advice with extreme dedication and interest to improve the quality of the manuscript; finally, for the grammar revision we used the MDPI service.
We think that the work done in these 6 months on this manuscript made it an adequate review for the aim of the special issue; highlighting the importance of nutrition and vitamins during physical activity and the importance that these micronutrients have for the well-being of the individual.
Reviewer 2 Report
The suggestions and corrections were accepted and the manuscript has been improved as suggested.
Author Response

(The authors gave the same response as above.)

Academic Editor Notes
A great implementation of description and discussion of topic is necessary. The authors should mark the novelty character of this paper respect to previous ones. I suggest to change the type of paper into "Perspective"
Thanks for the suggestions, in this latest version we have completed the corrections suggested by the reviewers. In particular, the first reviewer suggested that we shorten the introduction and carry out a profound grammar correction, the second was very satisfied and finally the third asked us to improve and further adjust the paragraph on the microbiome and grammar.
On our part, there was a commitment to modify and accept the advice with extreme dedication and interest to improve the quality of the manuscript; finally, for the grammar revision we used the MDPI service.
We think that the work done in these 6 months on this manuscript made it an adequate review for the aim of the special issue; highlighting the importance of nutrition and vitamins during physical activity and the importance that these micronutrients have for the well-being of the individual.
Reviewer 3 Report
The authors attempt to review the role of vitamin nutrition on muscle and heart health and the gut microbiota in athletes. Comments:
The manuscript requires extensive editing for error in language. Even the title contains errors.
The beginning overview section is long and contains basic information that seems unnecessary for a journal publication.
The microbiota section contains no insights about vitamin status and microbiota abundance and diversity. The effect of exercise on microbiota parameters has been shown in multiple published studies and systematic reviews.
Many of the proposed links between vitamin status and exercise outcomes are discussed without direct evidence of a causal relationship.
The manuscript overall does not provide valuable new insights that can be implemented in the field of sports nutrition.
Author Response

(The authors gave the same response as above.)

Academic Editor Notes
A great implementation of description and discussion of topic is necessary. The authors should mark the novelty character of this paper respect to previous ones. I suggest to change the type of paper into "Perspective"
Thanks for the suggestions, in this latest version we have completed the corrections suggested by the reviewers. In particular, the first reviewer suggested that we shorten the introduction and carry out a profound grammar correction, the second was very satisfied and finally the third asked us to improve and further adjust the paragraph on the microbiome and grammar.
On our part, there was a commitment to modify and accept the advice with extreme dedication and interest to improve the quality of the manuscript; finally, for the grammar revision we used the MDPI service.
We think that the work done in these 6 months on this manuscript made it an adequate review for the aim of the special issue; highlighting the importance of nutrition and vitamins during physical activity and the importance that these micronutrients have for the well-being of the individual.
Round 2
Reviewer 3 Report
The authors have made considerable improvements but errors still exist.
Title: Athletes'
Lines 182-214: References are needed to support these statements.
197-8: the unit of measure for vitamin A is incorrect.
235: replace "taken with food" with "found in plant food." Also, note that cholecalciferol is found in animal foods.
240-276: References are needed to support these statements.
251: Use the ug unit of measure instead of IU.
Vitamin K: Reference needed throughout.
426-7: "part two" is incorrect English. Phylotypes is the correct spelling.
Reference list is not adherent to guidelines.
References 146, 147 are not in English. Use references from English journals.
Reference 113, a review, is used often. It is preferred to cite the original research article.
596: use double quotes surrounding "athlete's heart."
597: remove "effective."
600: blood levels
611: "was" instead of is.
631: Athletes'
726: result instead of product.
730: remove "and prevention."
Author Response
Dear Editor,
thank you for the Reviewer’s Report about our manuscript entitled “The biological role of vitamins in athlete ‘s muscle, heart and microbioma”, submitted to International Journal of Environmental Research and Public Health (IJERPH) in the Special Issue "Healthy Nutrition and Physical Activity". We have appreciated the comments received by the Reviewer and yourself and have carefully re-considered them in preparing a new version of the manuscript.
A point-by-point response to the comments is attached below.
We believe that the manuscript is now significantly improved thanks to the Reviewer’s inputs.
We hope that the new version of the paper deserve publication on International Journal of Environmental Research and Public Health.
Best regards,
Prof. Dr. Olga Scudiero and Prof. Dr.Giulia Frisso
Point-by-point response.
Reviewer 3
The authors have made considerable improvements but errors still exist.
Title: Athletes'
Lines 182-214: References are needed to support these statements.
197-8: the unit of measure for vitamin A is incorrect.
235: replace "taken with food" with "found in plant food." Also, note that cholecalciferol is found in animal foods.
240-276: References are needed to support these statements.
251: Use the ug unit of measure instead of IU.
Vitamin K: Reference needed throughout.
426-7: "part two" is incorrect English. Phylotypes is the correct spelling.
Reference list is not adherent to guidelines.
References 146, 147 are not in English. Use references from English journals.
Reference 113, a review, is used often. It is preferred to cite the original research article.
596: use double quotes surrounding "athlete's heart."
597: remove "effective."
600: blood levels
611: "was" instead of is.
631: Athletes'
726: result instead of product.
730: remove "and prevention."
We totally agree with the corrections and thank you for the suggestions.
Following that, in accordance with the recommendations, we have made corrections.
Corrections are shown in green and in track change.
This manuscript is a resubmission of an earlier submission. The following is a list of the peer review reports and author responses from that submission.
Round 1
Reviewer 1 Report
A few recommendations:
- Key words should not duplicate words in the title as both are search fields
- English grammar/sentence structure is broken at times; example L 34 should be is considered; is L 37 a sentence?; L117 delete "the"; L118-122 is unclear as well as a run on sentence. Also, one sentence paragraphs should be avoided; L138 process should be Processes
My main criticism is that there is little new information presented in your review, i.e., most of your paper reads like an undergraduate text. Further, there are some inaccuracies such as L52: How many vitamins are there? I recommend reworking the review focusing on the novel information like the gut microbiome.
Reviewer 2 Report
Line 37 – To avoid physical damage? Please be clearer.
Line 38 – Why is “scenario” presented in italics?
Line 39 – remove “in” before “some processes.”
Line 39 – “sports practice.”
Please avoid semicolons in the abstract. The sentences must be clear, simple, and direct, avoiding those “text breaks” caused by semicolons.
The authors repeat “in particular” twice in the abstract in small text space. Please avoid such unnecessary repetitions. Choose synonyms.
Line 42 - Please reword the sentence, present vitamins in alphabetical order, and use the past tense, as the work has already been done: “The study was particularly focused on vitamins A, B6, D, E, and K, emphasizing…”
Line 45 - Keywords: please replace those already in the title.
Line 49 – Remove quotation marks from “micronutrients.”
Line 52 – Remove “are” before classified.
Line 65 – “And also” is redundant. Please reword.
Lines 94-95 – Please reword text as suggested in Line 42. The phrase “see table 3” became isolated with no proper connective.
Line 118 – Physical activity is indicated/recommended for all ages? Please be clearer.
Line 123-138 – The authors inverted the sentences. First, they showed their main research target. Then, they presented a reason for that choice. It has to be the opposite.
Line 123 – To finish that sentence, prefer using the impersonal form of writing “This review was focused on…”
Line 133 – Why is “state of the art” presented in italics?
Line 189 – Please chose a better word to replace “therefore.” It is too soon to start a conclusion about vitamin D.
Line 234 – The authors say that “according to recent studies” but indicate a single reference (twice in a row). Please double-check and provide the other references that they are talking about or reword the sentence.
Line 294 – Authors use quotation marks indiscriminately throughout the text. Please be more cautious and use them where they are indispensable only. Most of the time, it is not necessary to indicate common names largely described in the literature in quotation marks, as in the context of this study and for the audience of this journal, such compounds are not exceptional.
Line 354 – What is the source of the vectors used in Figure 1? It is clear that the image has been prepared using an image editor but was not completely edited. Some of them (e.g., banana, carrot) even appear to have their background not fully edited. Please double-check, provide a proper edition, and indicate the source (vector stock, freepik, etc.).
Line 470 – As in Figure, Figure 2 is a schematic representation too.
Line 517 – Allison et al, [114] are analysed?? Please use the same pattern for quotations: at the end of the sentence or right after the citation.
Line 635 – Please avoid starting a conclusion with “in recent years.” Ten years from now, this article will no longer be current, and the reader will not consider that this text is discussing recent years. The same is valid for “in conclusion,” as it is evident that this is a concluding section.
Line 647 –Please correct “evaluation.”
As a review article, the authors must include the proper references for the information they provide in each line in the tables, once they are not those who reported those results. All the physiological functions (Table 1), deficiency disorders (Table 2), and hypervitaminosis effects (Table 3) must have their references indicated in an additional column.
Also, titles for tables are too general. The authors must be specific, indicating that those results are related to humans (as described in Line 134), as much research on this topic is also performed using animal models, which sometimes do not reflect the same behavior in humans.
Reviewer 3 Report
Dear Author,
to my regret, I have to reject your manuscript. I wrote 3 pages of comments while reading through your first two, which means it needs to be completely rewritten if you want to publish.
Here are my comments:
34: proper is too informal, use adequate or satisfactory.
36: it is rarely high energy that is needed in sports, unless high performance sports. usually, high energy intake is the reason to do sport.
37: this sentence is amazing, and well put.
38: what scenario? Why is the word in italics? the scenario is not described.
39: remove "some"
40: Vitamins are "vital", they need to be consumed through diet.
Overall, the microbiome part is not well described/included in the abstract.
49: Whys is micronutrient in parethesis? Particluar is also not correct, specific may be a better word.
50: our bodies are also found in nature. Vitamins are not produced by humans, so we need to consume them by diet.
55: Blatantly false. Water soluble vitamins are stored, but they are readily excreted when ingested, so they do not tend to accumulate. B12 can be stored for about 6 months before symptoms occur, and C for a couple of weeks.
62: D is present in Shiitake. Also, D3, the active form, is only produced by our bodies if all necessary compounds are present in the necessary amounts. the compounds are shared with other pathways, like stress hormone and pro-inflammatory pathway. Also, low fat diet can cause issues for vitamin D, and even in countries with high sunlight and UV, hypovitaminosis D is common.
64: vitamins do not help to provide energy, they are involved in energy metabolism.
66: Vitamins do not prevent dysbiosis. If it would be that easy, it would be great.
77: In inflammatory processes, Vitamins C, D, and A are depleted, and cause hypovitaminosis that way.
79: only severe lack of vitamin causes these, moderate lack has more subtle signs, like more severe symptoms during illness and being more receptive for illness.
90: Parenthesis? better: hypovitaminosis.
But supplementation has been shown to induce some cancers: B6 and bladder cancer in men, and lung cancer with Vitamin A/beta-carotenes in smokers
Tables: not very well aligned, do not look "orderly"
112-114: Vitamin D is necessary for cell division and cell repair. Also: why mention covid in a paper on dysbiosis, physical activity, and diet?
121: Acute, strenuous activity causes transient dysbiosis, because of overexertion. Marathon-related diarrhea, for example.
123-125: sentence does not make sense dramatically.
126-138: no mention why these vitamins are important for digestion/dysbiosis
Overall the link between vitamins, physical activity, and dysbiosis is not well explained. Dysbiosis is not cured by vitamins, nor prevented. They may prevent inflammation on the whole body, which may be located in the gut,. in individuals, but that is not dysbiosis. And everything we eat alters the microbiota.
Overall, you do need input from either a gastroenterologist, a dietician, or both, if you want to publish this paper.